# Vaccine against *Streptococcus suis* Infection in Pig Based on Alternative Carrier Protein Conjugate

**DOI:** 10.3390/vaccines10101620

**Published:** 2022-09-27

**Authors:** Natalie Kralova, Hana Stepanova, Jan Gebauer, Adam Norek, Katarina Matiaskova, Monika Zouharova, Katerina Nedbalcova, Vladimir Babak, Rea Jarosova, Peter Makovicky, Ivana Kucharovicova, Bronislav Simek, Hana Plodkova, Tomas Pecka, Jan Matiasovic

**Affiliations:** 1Veterinary Research Institute, 621 00 Brno, Czech Republic; 2Department of Experimental Biology, Faculty of Science, Masaryk University, 611 37 Brno, Czech Republic; 3Department of Morphology, Physiology and Animal Genetics, Faculty of AgriSciences, Mendel University in Brno, 613 00 Brno, Czech Republic; 4Department of Histology and Embryology, Faculty of Medicine, University of Ostrava, 701 03 Ostrava, Czech Republic; 5State Veterinary Institute Jihlava, 586 01 Jihlava, Czech Republic

**Keywords:** *Streptococcus suis*, conjugate vaccine, OVA, CRM197, pig

## Abstract

*Streptococcus suis* is a serious pathogen in the pig industry with zoonotic potential. With respect to the current effort to reduce antibiotic use in animals, a prophylactic measure is needed to control the disease burden. Unfortunately, immunization against streptococcal pathogens is challenging due to nature of the interaction between the pathogen and the host immune system, but vaccines based on conjugates of capsular polysaccharide (CPS) and carrier protein were proved to be efficient. The main obstacle of these vaccines is manufacturing cost, limiting their use in animals. In this work, we tested an experimental vaccine against *Streptococcus suis* serotype 2 based on capsular polysaccharide conjugated to chicken ovalbumin (OVA) and compared its immunogenicity and protectivity with a vaccine based on CRM197 conjugate. Ovalbumin was selected as a cheap alternative to recombinant carrier proteins widely used in vaccines for human use. We found that the ovalbumin-based experimental vaccine successfully induced immune response in pigs, and the IgG antibody response was even higher than after immunization with capsular polysaccharide-CRM197 conjugate. Protectivity of vaccination against infection was evaluated in the challenge experiment and was found promising for both conjugates.

## 1. Introduction

*Streptococcus suis* (*S. suis*) is currently considered to be one of the most serious bacterial pathogens in the pig industry and is widespread worldwide [1,2]. More than 33 different serotypes of *S. suis* and *S. suis*-like isolates have been described to date [3,4], but serotypes 1/2, 2, 3, 7 and 9 are considered the most clinically and economically important. Serotypes 2 and 14, in particular, have remarkable zoonotic potential, with most human cases in Southeast Asia [1]. Despite the considerable economic and clinical importance of *S. suis*, a vaccine for the prevention of the disease in pigs is not yet commercially available, except for autogenous vaccines. Unfortunately, autogenous vaccines are limited in their use on farms of vaccine strain origin [2]. Nevertheless, significant advances in *S. suis* vaccination research have been achieved with experimental conjugate vaccines [5].

Capsular polysaccharide is an important factor of *S. suis* virulence [6]. Antibodies against capsular polysaccharide are considered protective against *S. suis* infection [7,8]. Unfortunately, the mammalian immune system responds to polysaccharide antigens in a T-independent manner, thus developing mainly IgM antibody response with weak or no IgG antibody production and no memory B-cell development [9,10]. However, it has been shown that conjugation of capsular polysaccharide to the carrier protein enables the induction of glycoconjugate-specific CD4+ lymphocyte development and subsequent B-cell isotype switch to IgG production and memory B- cell development [11]. Conjugate vaccines are thus a smart solution for outwitting the immune system to develop a T-dependent immune response against T-independent antigens.

Numerous different carrier proteins have been successfully tested or even used for commercial conjugate vaccine preparation. Among bacterial or recombinant proteins, cross-reactive material 197 (CRM197) is widely used for vaccine production (HibTiter against *Haemophilus influenzae* infection, Wyeth Laboratories, Havant, Great Britain; Prevnar against *Streptococcus pneumoniae*, Pfizer, New York, USA; Menveo against *Neisseria meningitidis*, Novartis Vaccines and Diagnostics, Basel, Switzerland; [12]). Other bacterial toxoids were experimentally tested with good results [5,13,14]. Although bacterial or recombinant proteins were used as carrier proteins with great success, the main obstacle is the manufacturing cost of its preparation, which is not viable for veterinary use. Luckily, other proteins can be used as carrier proteins as well; for example, KLH (keyhole limpet hemocyanin), BSA (bovine serum albumin), cBSA (cationic bovine serum albumin) and OVA (chicken ovalbumin) were tested as carriers for hapten immunization [15]. OVA was also tested in experiments with *Meningococcus* vaccination [14].

Because OVA is far less expensive than other carrier proteins, we decided to test it for an experimental vaccine against *S. suis* serotype 2 infection. Thus, in the present study, we compare the efficacy of pig vaccination with two different experimental conjugate vaccines, CPS-OVA and CPS-CRM197. The vaccine based on CPS-OVA conjugate could be economically affordable for commercial use in the pig industry, while CPS-CRM197 represents a standard type of conjugate, similar to the conjugates used in successful commercial vaccines for human use. Immunogenicity of both conjugates was first evaluated in a mouse model and proved to be immunogenic and protective against infection in pigs.

## 2. Materials and Methods

### 2.1. Bacterial Strain

*Streptococcus suis* isolate 10/18/Ji/830 (State Veterinary Institute Jihlava, Czech Republic) obtained from the lungs of a diseased pig was used for a vaccine preparation as well as for the experimental challenge. The isolate was identified as *S. suis* by biochemical tests and then by mass spectrometry (Bruker Microflex and software Maldi Biotyper 3.0, Database CD BTYP3.0–Library; Bruker, Billerica, MA, USA) using the MALDI TOF method (Matrix-Assisted Laser Desorption/Ionization–Time of Flight). The isolate was identified as serotype 2 by multiplex PCR according to Kerdsin et al. [16] with subsequent whole-genome sequence analysis and PCR-RFLP [17].

### 2.2. Capsular Polysaccharide Isolation

Isolation of *S. suis* capsular polysaccharide was performed with some modifications, as described by Elliott and Tai [18]. Briefly, colonies of *S. suis* serotype 2 field strain were inoculated in 10 mL of Todd Hewitt broth (THB, prepared according to the instructions of the manufacturer Hardy Diagnostic, Santa Maria, CA, USA) and incubated for 5–7 h at 37 °C with agitation of 220 rpm and subsequently incubated in 2 L of THB for 18 h at 37 °C with agitation of 180 rpm. A sample was taken for bacterial counts. The bacterial culture was washed once in DPBS and centrifuged (45 min at 3790 rcf and room temperature). The bacterial pellet was resuspended in 100 mL of 0.1 M glycine buffer (pH 9.2) containing 30 mg of crystalline egg white lysozyme (lysozyme from chicken egg white, 70,000 units/mg solid, Merck KGaA, Darmstadt, Germany) and 200 µL of 10% sodium azide. The suspension was incubated overnight at 37 °C with an agitation of 180 rpm. During incubation with lysozyme, streptococcal components, including crude capsular polysaccharides (cCPS), were released into the supernatant. After the incubation, the suspension with residual debris was centrifuged (40 min at 4630 rcf). In contrast to cCPS used as an antigen for western blot and ELISA, lipids and proteins were removed from the supernatant by phenol:chloroform extraction (UltraPure BufferSaturated Phenol, Thermo Fisher, Waltham, MA, USA; Chloroform, Erba Lachema s.r.o., Brno, Czech Republic), to obtain pure CPS (pCPS) used for the immunization of animals alone or conjugated to carrier proteins. The supernatant was mixed with an equal volume of phenol:chloroform for 30 min on a magnetic stirrer. The suspension was centrifuged twice at 4 °C, with low acceleration and 1290 rcf to remove organic phase-containing proteins and lipids. The nucleic acids were subsequently removed from the aqueous phase by precipitation with ethanol (25% *vol*/*vol*) and 0.1 M CaCl_2_. The capsular polysaccharides were then precipitated from the supernatant by increasing the ethanol concentration to 80% *vol*/*vol*. The suspension was incubated overnight at 4 °C and then centrifuged at 3970 rcf and 4 °C for 40 min. The precipitate was dissolved in PBS and dialyzed against PBS (SnakeSkin^TM^ Dialysis Tubing 3.5K MWCO, 16 mm, Thermo Fisher Scientific, Rockford, IL, USA). pCPS was purified by gel chromatography on a Superdex 16/600 200 pg column and eluted with PBS. The pCPS was sterilized by filtration through a 0.22 µm filter, and the presence of serotype 2 specific capsular polysaccharides was verified by western blot analysis using hyperimmune rabbit serum against *S. suis* serotype 2. The total polysaccharide concentration was measured with the phenol-sulfuric acid method. Pierce^TM^ BCA Protein Assay Kit (Thermo Fisher, Rockford, IL, USA) and spectrophotometer DS-11 FX (DeNovix, Wilmington, DE, USA) were used for protein and nucleic acid quantification, respectively.

### 2.3. CPS Conjugation on Carrier Proteins

The isolated pCPS was conjugated to the carrier protein OVA (albumin from chicken egg white, Merck KGaA, Darmstadt, Germany) and CRM197 (Fina BioSolution, Rockville, MD, USA) according to Goyette-Desjardins et al. [5] and Hermanson [19]. Briefly, the capsular polysaccharides were depolymerized using the Bandelin Electronic UW 2010 ultrasonic homogenizer (MS-72 probe, amplitude 75%, pulse 3.0 s/3.0 s, time 15 min (Bandelin Electronic GmbH & Co. KG, Berlin, Germany)). The depolymerized CPS was mixed with 0.046 M sodium periodate (Sigma-Aldrich, Merck KGaA, Darmstadt, Germany) on ice in the dark to obtain a final concentration of 1 mM NaIO_4_ in reaction solution. Mild periodate oxidation of CPS selectively transforms relatively unreactive hydroxyls of sialic acid into amine-reactive aldehydes. After one hour, the mild periodate oxidation of capsular polysaccharides was quenched by the addition of two equivalents of triethylene glycol (Merck KGaA, Darmstadt, Germany) and then dialyzed against water. The modified capsular polysaccharide was conjugated to the carrier proteins OVA or CRM197 in 0.1 M NaHCO_3_ (pH 8.1). Free aldehydes of the capsular polysaccharides interact with the proteins to form a Schiff base, which is then reduced with sodium cyanoborohydride (Sigma-Aldrich, Merck KGaA, Darmstadt, Germany). The mixture was sealed with parafilm and incubated with agitation at 37 °C for 2 days in the dark. After 2 days, sodium borohydride (Sigma-Aldrich, Merck KGaA, Darmstadt, Germany) was added to reduce the remaining aldehydes and to form a stable amine. To remove excess reductant from the CPS–carrier protein conjugates, the mixture was extensively dialyzed against PBS buffer (SnakeSkin^TM^ Dialysis Tubing 3.5K MWCO, 16 mm, Thermo Fisher Scientific, Rockford, IL, USA). The capsular polysaccharide conjugated to the carrier protein was then purified by FPLC. After the purification, the concentrations of total carbohydrates and proteins were measured using the phenol–sulfuric acid method and the Pierce^TM^ BCA Protein Assay Kit, respectively.

### 2.4. Vaccine Preparation

The antigens used for mouse immunization (DPBS, pCPS, OVA, CRM197, conjugate CPS-OVA and conjugate CPS-CRM197) were mixed with Complete Freund Adjuvant (CFA; Sigma-Aldrich, Merck KGaA, Darmstadt, Germany) for the first dose of the vaccine or with Incomplete Freund Adjuvant (IFA; Sigma-Aldrich, Merck KGaA, Darmstadt, Germany) for the second dose of the vaccine. The immunization dose used for both vaccinations was 0.1 mg of antigen in 0.2 mL of emulsion antigen:adjuvant 1:1.

The CPS-OVA and CPS-CRM197 conjugates were also used to prepare vaccines for pig immunization. The immunization dose per one animal was 0.25 mg of antigen in 2 mL of emulsion antigen:adjuvant 1:1. For pig immunization, the Montanide ISA 50 V2 (Seppic, Courbevoie, France) adjuvant was used. Homogenization was performed by repeated aspiration using a syringe and a needle.

### 2.5. Experimental Animals

The experiments on animals were approved by the committee of the Ministry of Agriculture, protocol numbers MZe 1921 and MZe 2254.

### 2.6. Vaccination

#### 2.6.1. Mice

Six groups of mice, each consisting of ten animals, were immunized with the following antigens and CFA: DPBS, CPS, OVA, CRM197, conjugate CPS-OVA and conjugate CPS-CRM197. Three weeks later, all animals were revaccinated with the same antigen, but IFA was used instead of CFA. Vaccine was applied subcutaneously. Two weeks after the second dose of a vaccine, all animals were euthanized and spleen and blood were collected for the analysis of antibody response.

#### 2.6.2. Pigs

Three groups of weaned piglets, each group consisting of ten animals, were used in the experiment at 28 days of age. After one week of housing, one group was immunized with the CPS-OVA vaccine and the second group with the CPS-CRM197 vaccine, both containing the ISA50V2 adjuvant, and the third control group remained not immunized. Three weeks after the first dose (D21), all vaccinated animals received the second dose of the same vaccine. Both vaccine doses were of 2 mL volume and were applied intramuscularly on the left side of the neck. On the day of immunization (D0), 5 mL of blood was collected from *vena jugularis*, and then on day 21 (D21), day 35 (D35) and day 42 (D42) after the first vaccination. D35 was the day of the challenge infection, while D42 was the end of the experiment.

### 2.7. Challenge

Two weeks after the second dose of vaccine (D35), pigs in all three groups were infected intraperitoneally [5] with 2 × 10^9^ CFU/2 mL of *S. suis* isolate 10/18/Ji/830. The bacterial inoculum was prepared by growing the *S. suis* strain on blood agar plates containing 5% sheep blood (LabmediaServis, Jaroměř, Czech Republic) overnight and then in THB for 4 h. Afterwards, the culture was washed twice in PBS and diluted in PBS to 1 × 10^9^ CFU/mL. The purity and the concentration of the final inoculum were confirmed by 10-fold dilutions plated on blood agar plates. The health status and body temperature of experimental animals were checked every day after challenge. Animals were euthanized on day 7 after the challenge, or earlier in the case of serious clinical outcomes of infection. Subsequently, an autopsy was performed and organs for quantitative bacteriology and histology were sampled.

### 2.8. Quantitative Bacteriology

Samples from the lungs, brain and spleen and swabs from the carpal joint and pericardium were taken for the bacteriological determination of *S. suis*. Approximately 0.060 g of tissue weighed to three decimal places was added to 1 mL PBS with 10 homogenization beads (2.3 mm zirconia/silica beads; Biospec, Bartlesville, OK, USA) and then homogenized using the MagNA Lyser (Roche, Basel, Switzerland) for 30 s. A dilution series was prepared for the homogenized samples. A volume of 100 μL of diluted sample was spread on blood agar (Trios Ltd., Prague, Czech Republic) and incubated overnight at 37 °C. Grown colonies were identified by MALDI-TOF MS (Bruker Microflex and software Maldi Biotyper 3.0, Database CD BTYP3.0–Library; Bruker, Billerica, MA, USA). *Streptococcus suis* serotype 2 was confirmed by PCR [16] and PCR-RFLP [17].

### 2.9. Enzyme-Linked Immunosorbent Assay (ELISA)

Blood samples from the mice were collected before the first vaccination (D0), before the second vaccination (D14) and 14 days after the second vaccination (D28). Blood was collected from the piglets before the first immunization dose (D0), before the second immunization dose (D21), before the infection challenge (D35) and on the day the pigs were euthanized (D42). The blood was centrifuged at 850 rcf for 10 min and serum was collected and frozen at −20 °C until further use. The level of antibodies in the serum of mice and piglets was determined by indirect ELISA using the cCPS antigen isolated from the challenge strain of *S. suis* serotype 2. An indirect ELISA against protein antigens was also used to confirm the immunization with the CPS-OVA and CPS-CRM197 conjugate vaccines. Briefly, the cCPS, OVA and CRM-197 antigens were diluted in 0.05 M carbonate–bicarbonate buffer (pH 9.6) to a final concentration of 3 µg/mL. The wells of the clear flat-bottomed microtiter plates (NUNC Maxisorp, Thermo Fisher Scientific, Roskilde, Denmark) were coated with 100 µL of respective antigens and incubated overnight at 4 °C. The plates were washed five times with a washing buffer, PBS containing 0.05% Tween 20 (PBST), using a semi-automatic microplate washer (BioTek Instruments Inc, Winooski, VT, USA) and blocked with 150 µL PBST containing 0.1% EDTA, 0.5% casein sodium salt, 10% sucrose and 0.01% sodium azide for 30 min. After the aspiration of the blocking buffer, the mouse or pig serum was 100× diluted in PBST containing 0.5% casein and incubated in wells for 1 h. After washing, the plates were incubated with horseradish peroxidase (HRP)-conjugated goat anti-mouse IgG, HRP-conjugated goat anti-swine IgG (1:30,000) or HRP-conjugated goat anti-swine IgM (1:10,000) (Bethyl Laboratories, Montgomery, TX, USA) for 1 h. The plates were washed five times with a washing buffer, and then 100 µL of 3,3′,5,5′-tetramethylbenzidine (TMB-Complete 2, TestLine Clinical Diagnostics s.r.o., Brno, Czech Republic) substrate was added to detect HRP. The enzyme reaction was stopped by the addition of 50 µL of 1 M H_2_SO_4_, and the absorbance was read immediately at 450 nm with the ELISA reader (Synergy H1, BioTek Instruments Inc, Winooski, VT, USA).

### 2.10. Western Blot

Samples of carrier proteins (OVA, CRM197), cCPS and conjugates were separated on 12.5% acrylamide gel by sodium dodecyl sulfate polyacrylamide gel electrophoresis (SDS-PAGE; 90 V/10 min, then 130 V/60 min, constant voltage) and stained with Coomassie Brilliant Blue R-250 Dye, stained with silver, or transferred on Immobilon-P Transfer PVDF Membrane (Millipore, Merck KGaA, Darmstadt, Germany) using the western blotting method. Briefly, the acrylamide gel was stained with Coomassie Brilliant Blue Dye for approximately 2 h and then left in a decolorizing solution (25% ethanol, 7% acetic acid) for about 30 min. For silver staining, the gel was fixed overnight in a solution of 50% methanol, 12% acetic acid and 3% glutaraldehyde. The next day, the gel was washed five times in distilled water and incubated in a silver solution (400 µL NH3, 700 µL 20% AgNO_3_ in 35 mL H_2_O) for 15 min. After washing, the gel was stained in the developer (0.15% formaldehyde 0.005% citric acid). The reaction was stopped by adding 7% acetic acid. Samples separated by SDS-PAGE were transferred to the PVDF membrane using an electroblotting instrument (Omni-bio, Brno, Czech Republic). The PVDF transfer membrane was activated in methanol, washed in distilled water and then incubated in a transfer buffer (25 mM Tris, 192 mM glycine and 15% methanol). Semi-dry transfer was performed at 360 mA for 90 min. The PVDF membrane was blocked overnight in DILUENT 13 (TestLine Clinical Diagnostics s.r.o., Brno, Czech Republic). The membrane was incubated either with homemade serum from rabbit immunized with *S. suis* serotype 2, anti-chicken egg albumin (OVA) monoclonal antibody (Sigma-Aldrich, Merck KGaA, Darmstadt, Germany), anti-Diphtheria toxin A subunit monoclonal antibody [8A4] (Abcam, Cambridge, UK) or mouse serum (dilution 1:1000) for at least 2 h with agitation. After washing in PBST, the membrane was incubated for 90 min with conjugated secondary antibody (HRP-conjugated anti-rabbit IgG 1:1000, HRP-conjugated goat anti-mouse IgG 1:1000; Bethyl Laboratories, Montgomery, TX, USA) in PBST with 0.5% casein. The membrane was washed three times and developed with 3,3′-diaminobenzidin-tetrahydrochlorid (Erba Lachema, Brno, Czech Republic) and H_2_O_2_ in PBS.

### 2.11. Cellular Immunity

Cellular immune response of pigs was evaluated at the end of the experiment (D42) by flow cytometry interferon gamma (IFNγ) detection after the re-stimulation of blood mononuclear cells with a specific antigen. The peripheral blood mononuclear cells (PBMCs) were collected from heparinized blood by gradient centrifugation using Histopaque (Sigma-Aldrich, Merck KGaA, Darmstadt, Germany) with a density of 1.077 g/mL. PBMCs were cultured in RPMI 1640 medium (Sigma-Aldrich, Merck KGaA, Darmstadt, Germany) supplemented with 10% fetal calf serum (Diagnovum, Ebsdorfergrund, Germany) and antibiotics (100,000 IU/L penicillin, 100 mg/L streptomycin; Sigma-Aldrich, Merck KGaA, Darmstadt, Germany) in microplates at 37 °C with an atmosphere supplemented to 5% of CO_2_. The cells were re-stimulated in vitro with antigen (CRM-197, OVA, cCPS) for 8 h. The cells only with medium were left as negative control samples. Subsequently, a protein transport inhibitor, brefeldinA (10 μg/mL, Sigma-Aldrich, Merck KGaA, Darmstadt, Germany), was added for the next 6 h. After incubation, cells were harvested and stained for surface markers CD3 and CD4. The following combinations of primary antibodies were used: anti-CD3 (clone BB23-8E6-8C8, PerCP-Cy5.5 conjugated, BD Biosciences, Franklin Lakes, NJ, USA) and anti-CD4 (clone 74-12-4, unconjugated; Washington State University, Pullman, WA, USA). Phycoerythrin-conjugated mouse isotype-specific (IgG2b) goat antiserum (Invitrogen, Thermo Fisher Scientific, Rockford, IL, USA) was used for unconjugated primary antibody visualization. After surface marker staining, the samples were fixed and cells permeabilized using the BD Cytofix/Cytoperm kit (BD Biosciences, Franklin Lakes, NJ, USA) and stained with anti-IFNγ antibody (Alexa Fluor 647-conjugated, CC302; Bio Rad, Hercules, CA, USA). Samples were acquired on an LSR Fortessa flow cytometer (BD Biosciences, Franklin Lakes, NJ, USA). For the data analyses performed in Diva software (BD Biosciences, Franklin Lakes, NJ, USA), doublets were excluded from analysis. CD4+ cells were defined from all CD3+ cells and the percentage of IFNγ-producing cells (IFNγ+) cells from the CD3+CD4+ population was evaluated. The cellular response of pigs was measured as a ratio of IFNγ-producing CD4+ cells in non-stimulated sample vs. sample stimulated with antigen (CRM-197, OVA or cCPS). The gating strategy is available in the Appendix A.

### 2.12. Histology

During necropsy, samples from the lungs, brain and spleen were collected. Histology was performed as described previously [20]. Briefly, tissue samples were embedded into the cryoprotective medium and frozen in pre-cooled n-heptane (Penta, Praha, Czech Republic) placed on dry ice. The tissue samples were then cut to a thickness of 5–10 μm on the cryostat (Leica Microsystems, CM 1900, GmbH, Wetzlar, Germany) at a temperature of −20 °C. The cuts of tissue were placed on slides, and the sections were allowed to dry at room temperature, fixed in pre-cooled acetone (Penta, Praha, Czech Republic) at −18 °C for 5 min and stained with hematoxylin-eosin (HE) and Gram staining.

#### 2.12.1. HE Staining

The slides were rehydrated in PBS for 3 min and then rinsed in a stream of tap water for 3 min until the cryoprotective medium was washed off. The slides were stained using Mayer’s hematoxylin (Penta, Praha, Czech Republic) for 30 s and then washed in tap water for 1 min; blue nuclei were rinsed in PBS for 20 s and then washed in tap water for 1 min. After that, the slides were dipped in 70% and 95% EtOH (Penta, Praha, Czech Republic) for 30 s and counter-stained in alcoholic eosin solution (Penta, Praha, Czech Republic) for 30 s. The slides were dehydrated through 2 changes of 95% and 3 changes of 100% EtOH for 15 s and cleared through 3 changes of xylene (Penta, Praha, Czech Republic) for 1 min each and cover-slipped.

#### 2.12.2. Gram Staining

The slides were incubated with crystal violet for 2 min, rinsed with Lugol (P-lab, Praha, Czech Republic) solution and incubated in Lugol solution for 3 min. The slides were then washed with Gram’s decolorizing solution, washed in distilled water and stained with carbolic fuchsine solution for 1 min, then washed, dried and dehydrated by ascending alcohol series, cleared with xylene and mounted.

### 2.13. Statistical Evaluation

Pig ELISA, body temperature and cellular immunity (OVA and CRM197 specific) data were analyzed using a two-way ANOVA with the factors of group and day in a repeated measures design with Geisser–Greenhouse corrections for the violation of the sphericity assumption, and with Tukey’s HSD and Dunnett’s post-hoc tests. Mice ELISA data were analyzed using a one-way ANOVA with Sidak’s test. The presence of bacteria in organ samples and cCPS-specific cellular immunity were analyzed with Fisher’s exact test. Data analysis was performed using statistical software Statistica 13.2 (StatSoft Inc., Tulsa, OK, USA) and package stats (R-project 4.1.2). *p* values ≤ 0.05 were considered statistically significant. Graphs were constructed using GraphPad Prism 9.0.0. (GraphPad Software, San Diego, CA, USA).

## 3. Results

### 3.1. Confirmation of CPS Conjugation to Carrier Proteins

Conjugation of the capsular polysaccharide to the carrier proteins was verified by band shift in Coomassie Brilliant Blue staining, silver staining and western blot with specific antibody reactions (Figure 1). Both Coomassie and silver-stained gel showed thicker bands of the conjugates and showed a shift in molecular weight from ~40 kDa to ~70 kDa (CPS-OVA) and a shift from ~58 kDa to ~260 kDa (CPS-CRM197). Western blot identified conjugates with protein-specific primary antibodies that were targeted by the secondary antibodies and visualized. The CPS-OVA conjugate reacted with anti-*S. suis* serotype 2 rabbit serum (Figure 1C) and anti-chicken egg ovalbumin monoclonal MAb (Figure 1D); similarly, the CPS-CRM197 conjugate reacted with anti-*S. suis* serotype 2 rabbit serum (Figure 1C) and anti-Diphtheria toxin A subunit MAb [8A4] (Figure 1E). Purified depolymerized pCPS bands were not visible in either Coomassie or silver staining, or in Western blot using anti-*S. suis* serotype 2 rabbit serum. This is due to the inability of this organic dye to bind to saccharide molecules, since it makes complexes mainly with basic amino acids in proteins. On the other hand, conjugates with the anti-*S. suis* serotype 2 rabbit serum are visible, indicating that CPS is covalently bound to carrier proteins.

### 3.2. Mice Antibody Response to Vaccination

To test whether our vaccine preparation induces an immune reaction, we first tested different variants of vaccine antigens in mice. We found that two weeks after the second vaccine dose, both conjugates induced IgG antibody production against the capsular polysaccharide antigen. Moreover, the CPS-specific antibody response was stronger in mice immunized with CPS-CRM197 than with CPS-OVA. As expected, mice immunized with carrier proteins OVA or CRM197 did not respond to cCPS at all. Neither mice immunized with pCPS nor the control group reacted to cCPS (Figure 2). To determine the levels of antibody in mouse serum, the ELISA assay with cCPS as an antigen was used. Serum from mice immunized with both conjugates induced highly significant responses to cCPS (*p* ≤ 0.001, one-way ANOVA) two weeks after the second vaccine dose, while the group of mice vaccinated with pCPS only, CRM197 and OVA proteins only, and the control group did not have an antibody immune response (Figure 3).

### 3.3. Pig Immune Response to Vaccination

Based on immune reaction to pCPS alone and CPS conjugates data obtained in the mice experiment, pigs were vaccinated only with CPS-OVA and CPS-CRM197 antigens.

The level of specific IgM and IgG antibodies against the crude capsular polysaccharide was measured by the ELISA assay in all pigs.

#### 3.3.1. IgM

In the CPS-OVA and CPS-CRM197 groups, the IgM serotype 2 specific antibody level was significantly increased (*p* ≤ 0.05, two-way ANOVA) at D21 (day of revaccination) compared to D0 (day of vaccination). In the control (not vaccinated) group, a low but statistically significant increase was seen from D35 (day of infection) compared to D0.

IgM values at individual time points differ significantly only between the control and the CPS-OVA group at time points D21 and D35 (Figure 4).

#### 3.3.2. IgG

In the CPS-OVA and CPS-CRM197 groups, the IgG serotype 2 specific antibody level was significantly increased (*p* ≤ 0.05; two-way ANOVA) from three weeks after the first dose (D21) and two weeks after the second dose (D35), respectively, compared to the day of immunization (D0) (Figure 5). In the control (not vaccinated) group, the significant increase in specific IgG level was found only after the infection challenge on D42 when compared to D0.

There was no difference between groups on D0, while on D21 and D35; IgG levels between all groups were significantly different, with the highest serotype 2 specific antibody levels in the CPS-OVA group. On D42, the only significant difference was between the control and CPS-OVA groups.

### 3.4. Cellular Immunity

#### 3.4.1. Cellular Response to cCPS

The cellular response of pigs was measured as a ratio of IFNγ-producing CD4+ cells in the non-stimulated sample vs. the sample stimulated with cCPS (Figure 6).

Although there seemed to be more IFNγ-producing CD4+ cells in vaccinated groups, especially in the CPS-OVA group, this difference was not statistically significant (*p* > 0.05, Fisher’s exact test) when compared to the control (not vaccinated) group.

In the CPS-OVA group, cCPS-specific stimulation was detected in only four animals. The specific response was detected in only three animals in CPS-CRM197 group, and the percentage of cCPS-specific CD4+IFNγ cells was lower in this group than in the CPS-OVA group. There was a low response to the cCPS or no response at all in the control group.

#### 3.4.2. Comparison of Cellular Immune Response to Carrier Proteins and cCPS

Cellular response was measured as a ratio of IFNγ-producing CD4+ cells in the non-stimulated sample vs. the sample stimulated with CRM197 or OVA (Figure 7).

Both conjugates induced more CD4+ cells responding to carrier proteins than cells responding to cCPS only.

Antigens OVA and CRM197 specifically stimulated only the CD4+ cells from animals immunized with matching conjugate in comparison to other groups (*p* ≤ 0.05, two-way ANO-VA) (Figure 7). CD4+ cells in the CPS-OVA group responded to OVA re-stimulation in six animals; however, this number of responders was not significantly higher (*p* > 0.05, Fisher’s exact test) than the number of responders to cCPS stimulation. In CPS-CRM197, CD4+ cells responded to CRM197 stimulation in nine animals and it was a significantly higher (*p* ≤ 0.05, Fisher’s exact test) number of responders than responders to cCPS stimulation. In the control group, only one animal showed a high response to both OVA and CRM197 antigen stimulation.

### 3.5. Bacteriology Examination of Organs

Most organ samples positive for the challenge strain were found in the control group, where five animals were culture-positive in at least one organ (Table 1).

In the CPS-OVA group, organ samples from two animals were culture-positive. Animal No. 12 had to be euthanized on day five post-infection (DPI5) and all sampled organs were culture-positive. In the second animal (No. 17), only lungs were culture-positive.

In the CPS-CRM197 group, only one animal was culture-positive and only on the swab from the pericardium.

A statistically significant (*p* < 0.05; Fisher’s exact test) difference in culture-positive samples was found between the CPS-CRM197 and both the control and the CPS-OVA groups.

Considering that animal No. 12, unlike the others in the CPS-OVA group, may have been extremely susceptible to infection due to some unknown circumstances, we recalculated statistical evaluation without this animal. In that case, the difference between the CPS-OVA and the control group became significant (*p* ≤ 0.05; Fisher’s exact test), and there was no significant difference between the CPS-OVA and the CPS-CRM197 group.

### 3.6. Clinical Symptoms of Infection

In the control non-vaccinated group, one animal (No. 8) was tilting its head on DPI2 and DPI3. Two other animals (No. 2 and No. 5) had to be euthanized on DPI6 because of a severe course of infection showing depression and incoordination. Additionally, animal No. 2 in this group developed cyanosis of the ears and petechial body hemorrhage (Figure 8). Similarly, one animal (No. 12) from the group vaccinated with CPS-OVA was euthanized on DPI5 due to depression and incoordination. Other animals in all groups did not show clear symptoms of the infection.

The body temperature did not significantly (*p* > 0.05, two-way ANOVA) change during the course of infection, although a slight elevation in rectal temperatures was seen from DPI1 to DPI4 after the infection in all groups.

### 3.7. Histology

Histological changes were found in animals positive for the *S. suis* challenge strain in internal organs. Most apparent was interstitial pneumonia (Figure 9) as a result of systemic infection and the presence of cocci in the brain (Figure 10). Representative pictures are shown here.

## 4. Discussion

*Streptococcus suis* infections in pigs are economically and clinically important for the pig industry [21]. Vaccination against the pathogen could be the way to reduce the infection pressure, especially when there is a current effort to reduce the use of antibiotics in livestock [2]. Unfortunately, induction of protective immunity against streptococcal infections is a quite difficult task. Due to the course of interaction of streptococcal pathogens with the mammalian immune system, it seems to be crucial to ensure a sufficient antibody response against the capsular polysaccharide to prevent a severe clinical outcome of infection [22]. Although immunization with the purified capsular polysaccharide did not induce an IgG antibody response in our mouse model or in the other study [23], the immunization with the conjugate vaccines stimulated the production of the CPS-specific antibody in the mouse model. This confirms that the free polysaccharide is a very weak inducer of the IgG immune response [5,24], especially the polysaccharide from *S. suis* serotype 2 [25,26,27,28,29].

Research on conjugate-based vaccines for human use, namely, the *Streptococcus pneumoniae* vaccine (Prevnar, Pfizer), opens up opportunities to evaluate conjugate-based vaccines for the immunization of pigs against *S. suis* infection. Despite the success with *S. suis* vaccine construction reached [5], the vaccine against *S. suis* is still not available on the market. One of the important parameters for the use of the vaccine in animals is its price. In the case of the conjugated vaccine, the cost of the carrier protein, usually a recombinant protein, is limiting. Thus, we tested chicken ovalbumin as a cost-effective alternative to the commonly used CRM197 for its usability for the preparation of a vaccine against *S. suis*. In our experiment, the immunization of mice with the CPS-OVA or CPS-CRM197 conjugates successfully induced CPS-specific IgG production. These data encouraged us to perform a vaccination and challenge experiment in pigs. We found that in pigs, the CPS-OVA conjugate induced an even higher level of CPS-specific IgG than the CPS-CRM197 conjugate did. In light of the results with mice immunization and the use of CRM197 in commercial vaccines, the higher antibody response to CPS-OVA than to CPS-CRM197 in pigs was a surprising but welcome result.

Induction of CPS-responding CD4+ T-cells is considered crucial for the development of IgG-producing plasma cells and long-lived T-cell memory to polysaccharide antigen [5,26]. However, based on the nature of antigen processing within the mammalian immune system, the CPS alone as a T-independent antigen is not able to induce CPS-specific CD4+ T cells [11,30]. To overcome this problem, the concept of conjugated vaccines has been expanding in recent decades. In our experiment, we found that some pigs vaccinated with CPS-OVA or CPS-CRM197 conjugates developed CD4+ T-cells responding to CPS stimulation. However, this result was not significantly different from the control group (not vaccinated) because of high individual variability in both vaccinated groups. Although only some animals from both vaccinated groups developed CD4+ T-cells responding to CPS antigen, this contrasts with IgG production, where all vaccinated animals developed a CPS specific IgG antibody response. On the other hand, CD4+ T-cells responded much better to carrier proteins than to CPS, so it reveals the potential for further conjugate development [9,10,14]. It is possible that the CPS present in conjugates itself impairs CD4+ development [26].

Similar to the results obtained by Elliot et al. [27], the immunization results encouraged us to conduct a subsequent challenge of pigs. Although significantly lower than both vaccinated groups, control animals also developed some level of antibodies specific to CPS when measured one week after infection, demonstrating immune response to infection, although *S. suis* serotype 2 was reported as a weak antibody inducer [29]. The animals with the lowest level of CPS-specific IgG and IgM from the control group were positive for *Streptococcus suis* infection strain in lungs, brain and spleen, which corresponds to previous data correlating antibody levels and severity of infection [26]. On the other hand, the pig from the CPS-OVA group that was euthanized on day five post-infection produced CPS-specific IgG (on an average level of the group) on the day of infection, but all of its organ samples were culture-positive for the challenge strain. We did not identify the reason for such exceptional sensitivity to the challenge strain. Apart from this one animal, other individuals in the CPS-OVA group were well protected against the infection, similar to animals vaccinated with the CPS-CRM197 conjugate. This result is comparable to previous experiments with other conjugated vaccines against *S. suis* [5]. Taken together, CPS-OVA is promising as an alternative to the CRM197 carrier protein for the further development of vaccines against *S. suis* infection in pigs. With respect to experience with inactivated *S. suis* vaccines [2,31] and limited cross-protectivity of CPS from different *S. pneumoniae* serotypes (Prevnar, Pfizer), we can speculate that the efficient conjugated vaccine against *S. suis* infection in pigs could be based on conjugates containing CPS from most clinically and economically important serotypes.

## Figures and Tables

**Figure 1 vaccines-10-01620-f001:**
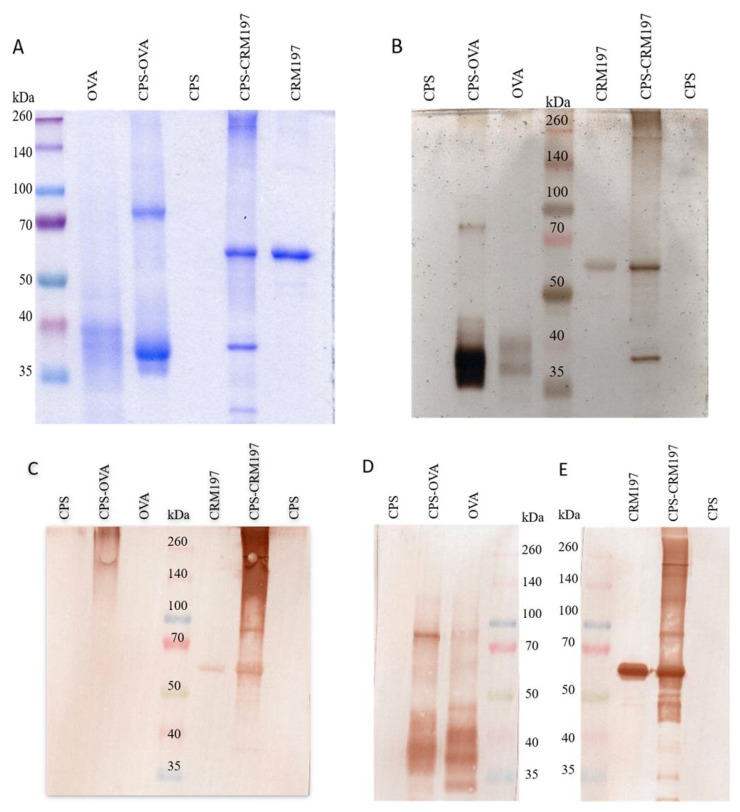
Confirmation of CPS conjugation to carrier proteins. Presence of new bands documents conjugation of CPS having various molecular weight to carrier proteins. Coomassie brilliant blue staining, silver staining and western blot; 12.5% acrylamide gels were stained after SDS-PAGE with Coomassie Brilliant Blue R-250 Dye (**A**), or stained with silver (**B**), or samples were transferred after SDS-PAGE to PVDF membrane and developed with anti-S. suis serotype 2 rabbit serum (**C**), anti-chicken egg albumin (ovalbumin) monoclonal antibody (**D**) or anti-Diphtheria toxin A subunit monoclonal antibody [8A4] (**E**). For SDS-PAGE, the marker Spectra Multicolor Broad Range Protein Ladder (Thermo Scientific) was used.

**Figure 2 vaccines-10-01620-f002:**
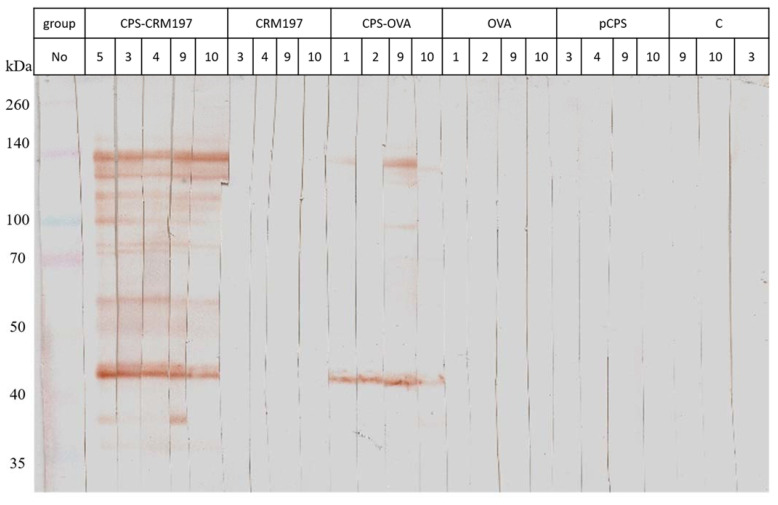
Mice antibody response to vaccination. The immunogenicity of antigens was tested using serum from different groups of immunized mice (CPS-CRM197, CRM197, CPS-OVA, OVA, pCPS and control groups). Immunoblot. Crude CPS sample was loaded on 12.5% acrylamide gel, separated by SDS-PAGE and transferred to the PVDF membrane. The numbers on the second line identify individual animals.

**Figure 3 vaccines-10-01620-f003:**
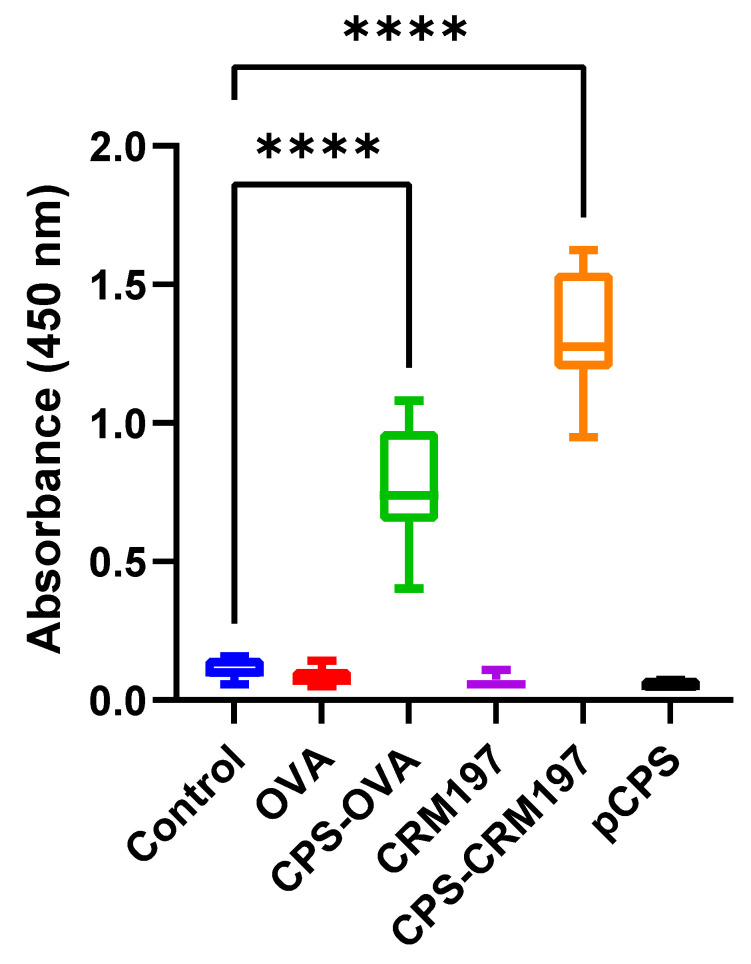
ELISA detection of mice anti-cCPS antibody (IgG). Total antibody levels against cCPS in mouse serum two weeks after the second vaccine dose. ELISA plates were coated with *S. suis* serotype 2 capsular polysaccharide and incubated with mice serum (dilution 1:100). Total levels of IgG were measured for all six groups (n = 10 each) at 450 nm. Both mice groups immunized with conjugates developed high antibody response, significantly higher (****, *p* ≤ 0.0001) than the control (not vaccinated), OVA, CRM197 or pCPS groups.

**Figure 4 vaccines-10-01620-f004:**
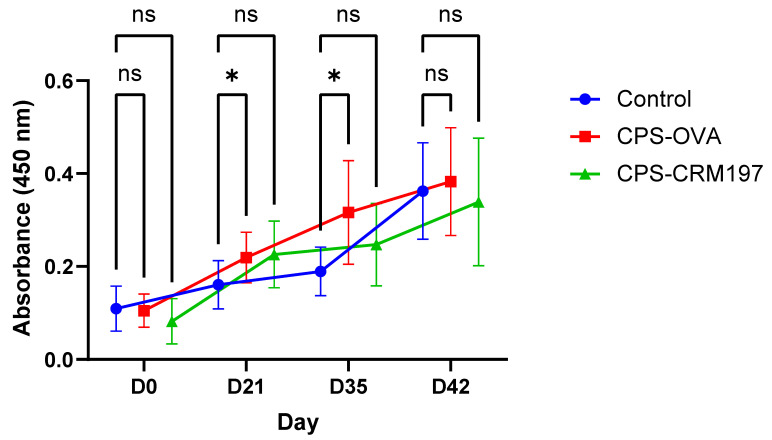
ELISA detection of porcine anti-cCPS antibodies (IgM). Total IgM levels against cCPS in pig serum. ELISA plates were coated with *S. suis* serotype 2 capsular polysaccharide and incubated with pig serum (dilution 1:100). IgM levels were measured for CPS-OVA, CPS-CRM197 and control (not vaccinated) groups (n = 10 each) on the day of vaccination (D0), the day of revaccination (D21), the day of infection (D35) and at the end of the experiment (D42) at 450 nm. The CPS-OVA group induced significantly higher antigen-specific IgM response (*, *p* ≤ 0.05) than the control group on D21 and D35.

**Figure 5 vaccines-10-01620-f005:**
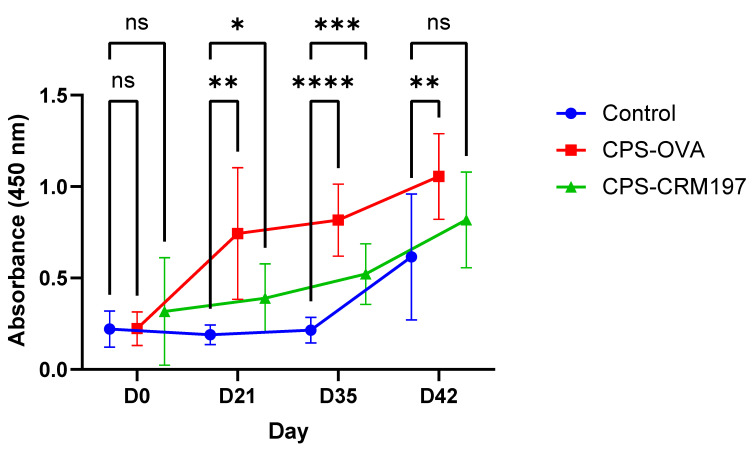
ELISA detection of porcine anti-cCPS antibodies (IgG). Total levels of CPS-specific IgG in pig serum. ELISA plates were coated with *S. suis* serotype 2 capsular polysaccharide and incubated with pig serum (dilution 1:100). IgG levels were measured for CPS-OVA, CPS-CRM197 and control (not vaccinated) groups (n = 10 each) on the day of vaccination (D0), the day of revaccination (D21), the day of infection (D35) and at the end of the experiment (D42) at 450 nm. In comparison to the control group, there was a significantly higher response in CPS-OVA (**, *p* ≤ 0.01) and CPS-CRM197 (*, *p* ≤ 0.05) groups on D21. There was also a significant response in CPS-OVA (****, *p* ≤ 0.0001) and CPS-CRM197 (***, *p* ≤ 0.001) groups on D35. On D42, the only significant difference (*p* ≤ 0.01) from the control group was observed in the CPS OVA group.

**Figure 6 vaccines-10-01620-f006:**
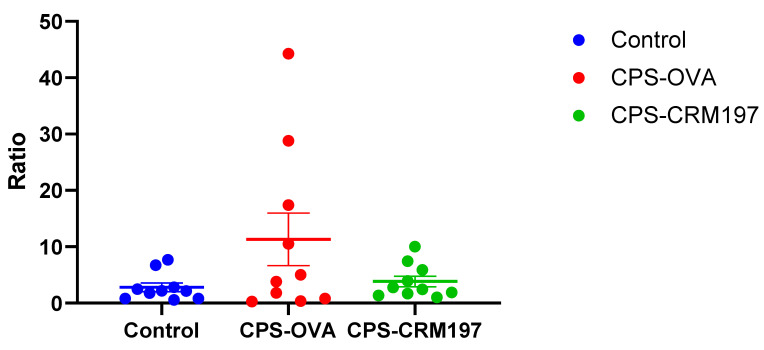
Cellular immune response to cCPS in vitro re-stimulation. Flow cytometry. Isolated PBMCs were re-stimulated in vitro with cCPS and subsequently stained for surface markers (CD3, CD4) and intracellular IFNγ. The ratio of IFNγ producing cells was analyzed from all CD3+CD4+ cells. The data are expressed as the ratio calculated: percentage of CD4+IFNγ+ from the cCPS-stimulated sample/percentage of CD4+IFNγ+ from the un-stimulated sample. Data are presented for individual animals from (not vaccinated) control, CPS-OVA- and CPS-CRM197-immunized pigs. The number of responders was not statistically different when both groups of conjugated-vaccinated animals were compared to controls (*p* > 0.05, Fisher’s exact test).

**Figure 7 vaccines-10-01620-f007:**
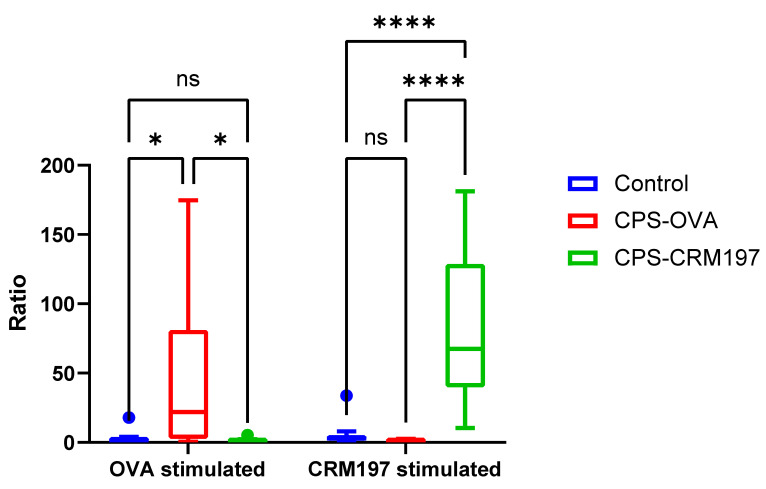
Cellular immune response to OVA or CRM197 in vitro re-stimulation. Flow cytometry. Isolated PBMCs were re-stimulated in vitro with OVA or CRM197 and subsequently stained for surface markers (CD3, CD4) and intracellular IFNγ. The ratio of IFNγ producing cells was analyzed from all CD3+CD4+ cells. The data are expressed as the ratio calculated: the percentage of CD4+IFNγ+ from OVA (or CRM197)-stimulated sample/percentage of CD4+IFNγ+ from the un-stimulated sample. Data are presented for individual animals from control, CPS-OVA- and CPS-CRM197-immunized pigs. In comparison to the control and CPS-CRM197-vaccinated groups, there was a significant response in the CPS-OVA group when cells were re-stimulated with OVA (*, *p* ≤ 0.05). Similarly, the CPS-CRM197-vaccinated group showed an extremely significant response when compared to the control or CPS-OVA-vaccinated groups (****, *p* ≤ 0.0001).

**Figure 8 vaccines-10-01620-f008:**
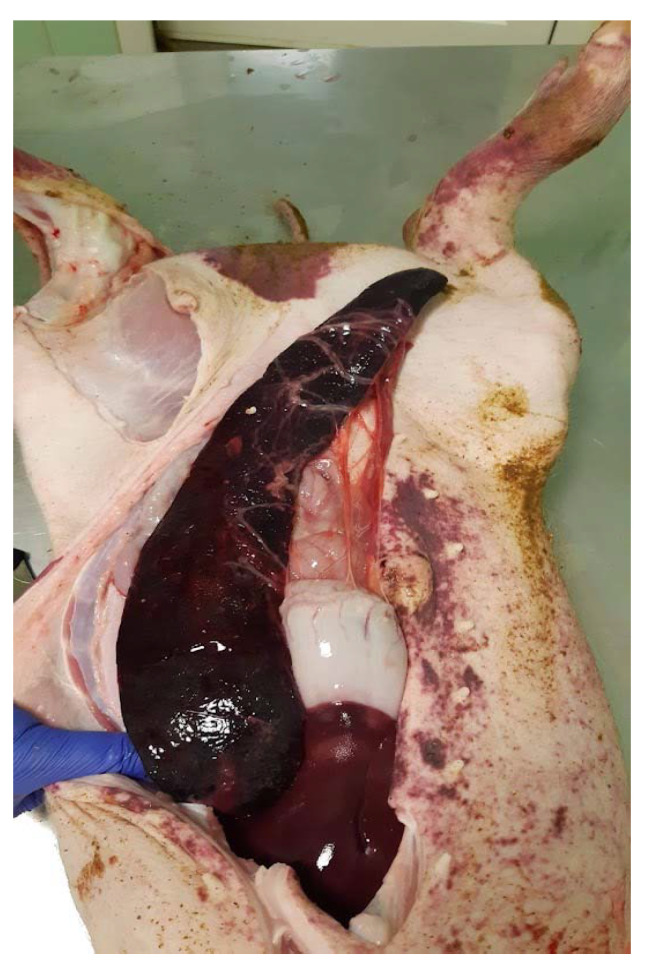
Severe clinical outcomes of *S. suis* infection. Animal No. 2 in the control group developed serious clinical outcomes of *S. suis* infection. Here, petechial body hemorrhage and splenomegaly are shown.

**Figure 9 vaccines-10-01620-f009:**
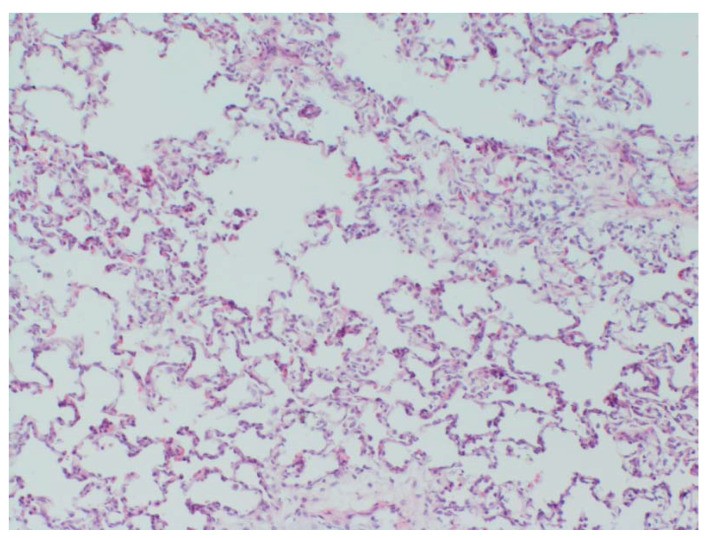
Interstitial pneumonia in animal No. 2. Lungs with increasing round cellular content in alveolar septa and in some places groups of cells in alveolar lumen are present. HE staining, 4×.

**Figure 10 vaccines-10-01620-f010:**
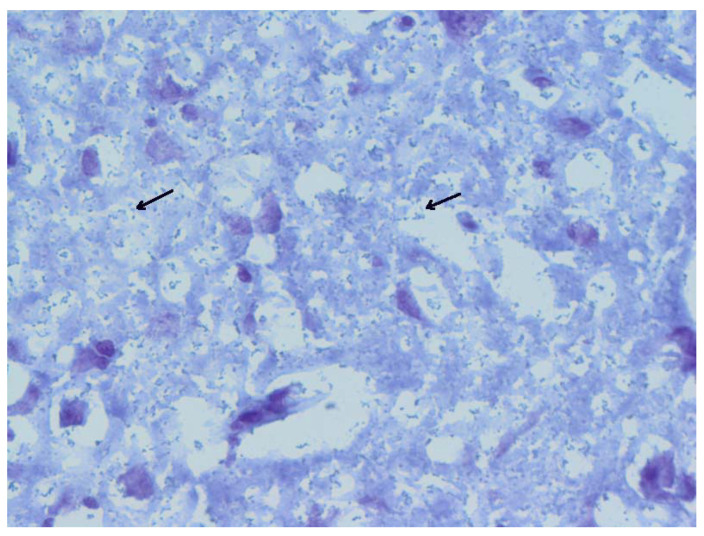
Detailed view of brain with mass of miniature cocci bacteria (tiny blue dots, black arrows pointing to examples) in animal No. 5. Gram staining, 40×.

**Table 1 vaccines-10-01620-t001:** *S. suis* serotype 2 colony number per gram of tissue.

Group	No	Lungs	Pericardial Swab	Brain	Spleen	Carpal Joint Swab
Control	1	0	0	0	0	0
	2	1.4 × 10^3^	0	3.6 × 10^4^	6.6 × 10^2^	0
	3	0	0	0	0	0
	4	1.8 × 10^3^	0	0	0	0
	5	4.7 × 10^5^	0	5.1 × 10^3^	2.8 × 10^7^	2 × 10^7^
	6	0	0	0	0	0
	7	0	0	0	0	4
	8	6.0 × 10^3^	0	1.6 × 10^2^	8.7 × 10^2^	0
	9	0	0	0	0	0
	10	0	0	0	1.6 × 10^2^	0
CPS-OVA	11	0	0	0	0	0
	12	2.1 × 10^3^	2 × 10	1.5 × 10^3^	9.5 × 10^4^	5.3 × 10^4^
	13	0	0	0	0	0
	14	0	0	0	0	2
	15	0	0	0	0	0
	16	0	0	0	0	0
	17	7.8 × 10	0	0	0	0
	18	0	0	0	0	0
	19	0	0	0	0	0
	20	0	0	0	0	0
CPS-CRM197	21	0	0	0	0	0
	22	0	0	0	0	0
	23	0	0	0	0	0
	24	0	5	0	0	0
	25	0	0	0	0	0
	26	0	0	0	0	0
	27	0	0	0	0	0
	28	0	0	0	0	0
	29	0	0	0	0	0
	30	0	0	0	0	0

## Data Availability

Not applicable.

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
