# Peer review of "Vaccine against Streptococcus suis Infection in Pig Based on Alternative Carrier Protein Conjugate"

_vaccines, 2022, doi:10.3390/vaccines10101620_

Round 1
Reviewer 1 Report
The manuscript "Vaccine against Streptococcus suis infection in pig based on alternative carrier protein conjugate" describes a potential cost-effective S. suis vaccine using conjugation of capsular polysaccharide (CPS) with chicken ovalbumin (OVA). Although CPS-OVA induced lower IgG titre in mice compared to the CPS-CRM197 (commercial control), the immunised pigs with CPS-OVA showed higher serum IgG titre and CD4+ T-cell responses.
The manuscript is clear and describes details, especially Materials and Methods. However, because of the depth of Materials and Methods, overall Discussion is relatively weak, especially T-cell response.
Excluding the unclear reason behind the mouse study, the reviewer wants to ask the authors why measuring IFNg-production CD3+CCD4+ population instead of ELISpot. Additionally, S. suis is intracellular bacteria, implying the importance of CD8+ T cells. However, the author did not measure CD8+ T-cell responses such as ICSs and subpopulation of CD8+ cells before and after immunisation and challenge.
Author Response
Reviewer 1
Comments and Suggestions for Authors
The manuscript "Vaccine against Streptococcus suis infection in pig based on alternative carrier protein conjugate" describes a potential cost-effective S. suis vaccine using conjugation of capsular polysaccharide (CPS) with chicken ovalbumin (OVA). Although CPS-OVA induced lower IgG titre in mice compared to the CPS-CRM197 (commercial control), the immunised pigs with CPS-OVA showed higher serum IgG titre and CD4+ T-cell responses.
The manuscript is clear and describes details, especially Materials and Methods. However, because of the depth of Materials and Methods, overall Discussion is relatively weak, especially T-cell response.
Excluding the unclear reason behind the mouse study, the reviewer wants to ask the authors why measuring IFNg-production CD3+CCD4+ population instead of ELISpot. Additionally, S. suis is intracellular bacteria, implying the importance of CD8+ T cells. However, the author did not measure CD8+ T-cell responses such as ICSs and subpopulation of CD8+ cells before and after immunisation and challenge.
Response: Thank you for your comments. The importance of mice immunization was on testing whether conjugates prepared in our laboratory are immunogenic and also to test the immune response to CPS alone. Results obtained on mice enabled us to reduce number of experimental groups in pig experiment to just three groups (not vaccinated control, immunization with CPS-OVA and immunization with CPS-CRM197).
We preferred to detect IFNg production measurement by flow cytometry because of direct recognition of white blood cell CD3+CD4+ subpopulation (T-helper cells; see gating strategy in Supplement 1). Based on our experience, the sensitivity of this method is even better than the ELISpot.
The goal of conjugate vaccines is the development of T-dependent type of immune response to T-independent antigens (the CPS in our case). Our experiment was thus focused on immunogenicity of CPS-OVA conjugate in the terms of CPS specific IgG production and CD3+CD4+ CPS specific T cells development. We try to improve T-cell discussion (lines 598-602).
Although there is evidence, that S. suis is able to invade cells and survive intracellularly (references please see below), S. suis is still recognized as an extracellular pathogen. As the main protective factor against infection and disease development is considered the antibody-enhanced phagocytosis (opsonophagocytosis). Because our work was focused on development of CPS specific IgG antibody and CD3+CD4+ CPS specific T cells after vaccination and because of yet not clear importance of cellular immunity for protection against S. suis infection, we do not measure presence of antigen-specific CD8+ cells.
Vanier G, Segura M, Gottschalk M. Characterization of the invasion of porcine endothelial cells by Streptococcus suis serotype 2. Can J Vet Res. 2007 Apr;71(2):81-9. PMID: 17479770; PMCID: PMC1829181.
Benga L, Friedl P, Valentin-Weigand P. Adherence of Streptococcus suis to porcine endothelial cells. J Vet Med B Infect Dis Vet Public Health. 2005 Nov;52(9):392-5. doi: 10.1111/j.1439-0450.2005.00880.x. PMID: 16283918.
Benga L, Goethe R, Rohde M, Valentin-Weigand P. Non-encapsulated strains reveal novel insights in invasion and survival of Streptococcus suis in epithelial cells. Cell Microbiol. 2004 Sep;6(9):867-81. doi: 10.1111/j.1462-5822.2004.00409.x. PMID: 15272867.
Holt, M. E., Enright, M. R., Alexander, T. J. Immunisation of pigs with live cultures of Streptococcus suis type 2. Res Vet Sci 1988, 45(3), 349-352.
Blouin, C., Higgins, R., Gottschalk, M., Simard, J. Evaluation of the antibody response in pigs vaccinated against Streptococcus suis capsular type 2 using a double-antibody sandwich enzyme-linked immunosorbent assay. Can J Vet Res 1994, 58(1), 49-54.
Reviewer 2 Report
This paper presents a vaccine against Streptococcus suis infection in pigs based on capsular polysaccharide (CPS) conjugated to chicken ovalbumin (OVA).
The study is interesting because it presents an easy and cheap way to produce a vaccine for S. suis infection in pigs. The introduction, methodology, and discussion are well done; however, from my perspective, some minor issues that need to be explained.
My comments are
- Line 418 “In the control group,..” What has been used to vaccinate the pigs in the control group? With pCPS? Please, mention in line 418 and the legend of figure 4.
- Lines 415-421 “IgM” A positive control sera were tested that increased from D0 to D42. However, I missed negative control sera from unvaccinated pigs or vaccinated with unrelated antigens. Sera were then kept unchanged from D0 to D42. Did you test such sera?
- Lines 435 “In the control group,..”. Just as above. Please, mention in line 435 and the legend of figure 5.
- Lines 431-440 “IgG” I missed negative control sera from unvaccinated pigs or vaccinated with unrelated antigens. Sera were then kept unchanged from D0 to D42.
- Lines 513-514....” Considering that animal No. 12,...” What was the level of CPS-specific IgG in the pig No. 12? Did that pig produce antibodies against CPS? and if that pig produce CPS-specific antibodies why they did not protect it from the infection? ..Could be something to do with the dosage used for the challenge that maybe is too high?.
- Line 587 “.., CD4+ T-cell responded much better to carrier proteins that to CPS” Probably, CPS is not a good inducer of CD4+ T-cells. Is any information available from other studies about this that you could comment on or compare with your results? It has been described in some articles that CPSs are generally considered poorly immunogenic since they are unable to recruit T cell help for B cell functions. Could please you comment on these papers in your discussion?
Mond, J.J.; Kokai-Kun, J.F. The multifunctional role of antibodies in the protective response to bacterial T cell-independent antigens. In Specialization and Complementation of Humoral Immune Responses to Infection;Manser, T., Ed.; Springer: Berlin/Heidelberg, Germany, 2008; Volume 319, pp. 17–40.
Goyette-Desjardins G, Calzas C, Shiao TC, Neubauer A, Kempker J, Roy R, Gottschalk M, Segura M. Protection against Streptococcus suis Serotype 2 Infection Using a Capsular Polysaccharide Glycoconjugate Vaccine. Infect Immun. 2016 Jun 23;84(7):2059-2075. doi: 10.1128/IAI.00139-16.
- Line 599 “..the pig from the CPS-OVA group that was euthanized on day five post-infection produced CPS-specific IgG (on an average level of the group) on the day of infection..” Is it known if CPS can generate different types of antibodies against different sugar compositions on CPS and if some of them provide more protection than others? Is any data from other studies about that that could explain the results? It seems that sialic acid in the CPS may interfere with the generation of anti-CPS Ab responses. Could you please comment on this in your paper?
Calzas C, Taillardet M, Fourati IS, Roy D, Gottschalk M, Soudeyns H, Defrance T, Segura M. Evaluation of the Immunomodulatory Properties of Streptococcus suis and Group B Streptococcus Capsular Polysaccharides on the Humoral Response. Pathogens. 2017 Apr 20;6(2):16. doi: 10.3390/pathogens6020016.
- Line 606 “CPS-OVA is promising..” There are several CPS serotypes..how could an ideal vaccine be? A cocktail of the different CPS serotypes or just serotype 2? Could you please comment on this in your discussion?
Author Response
Dear Editor,
We would like to thank very much to you and to both reviewers for giving us constructive comments and suggestions enabling to increase quality of our manuscript. We have carefully revised the manuscript and made revisions which are highlighted in yellow. Detailed point-by-point responses to the specific questions please find within your and reviewers comments. We amend one Supplement and three more citations.
On behalf of all co-authors,
Ján Matiašovic
-------------------------------------------
Dear Dr. Matiašovic,
Hope this message finds you well.
For all Western blot figures, please include densitometry readings/intensity ratio of each band. In addition, please include the whole blot (uncropped blots) showing all the bands with all molecular weight markers on the Western in the Supplemental Materials.
We look forward to hearing from you soon.
Kind regards,
Ms. Charlotte Chen
Section Managing Editor
Response: Thank you for this suggestion. We add original scans of gels and western blots to Supplementary Materials as a Supplement_S2.docx.
Densitometry of western blot is not always straightforward. Compared to Coomassie-stained SDS-PAGE gels where is linear dependence between the band intensity and protein concentration, western blot often elicits nonlinearity and can reach saturation. Despite this fact western blots are usually followed by densitometry graphs in cases where proper technical controls can be used. On the other hand, we believe that it is not applicable in our case. The reason is that we do not measure intensity of one particular band but we used western blots to document antibody response to antigens consisting from many bands of different molecular weight.
The gel on Figure 1 A and B documented CPS conjugation to carrier proteins. The result of successful conjugation is band shift and presence of new bands in lines with conjugates. On Figures 1 C, D and E we show the ability of conjugates to bind specific antibody.
On Figure 2 we demonstrate immunogenicity of different antigens to induce antibody response to CPS in mice. Because of presence of bands with various molecular weight in antigen used for western blot, it is hard to quantify intensity of antibody response by densitometry. To quantify this antibody response, we used more accurate ELISA method (Figure 3).
We improve Figure caption of Figures 1 and 2 to better explain what they represent.
----------------------------------------------------------------
Reviewer 2
Comments and Suggestions for Authors
This paper presents a vaccine against Streptococcus suis infection in pigs based on capsular polysaccharide (CPS) conjugated to chicken ovalbumin (OVA).
The study is interesting because it presents an easy and cheap way to produce a vaccine for S. suis infection in pigs. The introduction, methodology, and discussion are well done; however, from my perspective, some minor issues that need to be explained.
My comments are
- Line 418 “In the control group,..” What has been used to vaccinate the pigs in the control group? With pCPS? Please, mention in line 418 and the legend of figure 4.
Response: Thank you for this comment and we are sorry for not clear group description. We clarify, that the control group was not vaccinated with any kind of antigen (line 188, also on line 418).
- Lines 415-421 “IgM” A positive control sera were tested that increased from D0 to D42. However, I missed negative control sera from unvaccinated pigs or vaccinated with unrelated antigens. Sera were then kept unchanged from D0 to D42. Did you test such sera?
Response: The paragraph describing IgM results was probably written not clearly, we are sorry for that. In fact, we had no positive control sera. The control group was the group of animals, which were not vaccinated. The only compared sera were collected from this control group and groups vaccinated with CPS-OVA and CPS-CRM197. We add this information on line 188 and 418 and to figure caption of Figures 3, 4, 5 and 6 to clarify the information.
The IgM is known to be less specific than IgG. The low, but significant increase of IgM from D0 to D35 may be explained by cross-reactivity of IgM raised against other natural occurring bacteria, for example a commensal Streptococci, to the S. suis serotype 2 CPS. This explanation is supported by the IgG data, where the control group did not show any elevation of antibody specific to serotype 2 CPS until D42, a week after challenge at D35 (Figure 5; time point D42) thus indicating the antibody response to serotype 2 CPS occurred in control group only after infection (discussed on lines 593-596).
- Lines 435 “In the control group,..”. Just as above. Please, mention in line 435 and the legend of figure 5.
Response: Please see previous and following response.
- Lines 431-440 “IgG” I missed negative control sera from unvaccinated pigs or vaccinated with unrelated antigens. Sera were then kept unchanged from D0 to D42.
Response: Thank you for this note. The situation is the same as for IgM. In fact, we had no positive control sera. The control group was the group of animals, which were not vaccinated. The only compared sera were collected from this control group and groups vaccinated with CPS-OVA and CPS-CRM197. We add this information on line 188 and 435 and to figure caption of Figures 3, 4, 5 and 6 to clarify the information.
- Lines 513-514....” Considering that animal No. 12,...” What was the level of CPS-specific IgG in the pig No. 12? Did that pig produce antibodies against CPS? and if that pig produce CPS-specific antibodies why they did not protect it from the infection? ..Could be something to do with the dosage used for the challenge that maybe is too high?.
Response: It is very interesting issue. We discuss it on lines 600-605. As mentioned in Discussion, the animal No. 12 developed antibody response to CPS on average level of CPS-OVA group. We were not able to identify, why this animal was not protected against infection, while other animals within CPS-OVA group were protected. On the other hand, no vaccination and resulting immune response is 100% protective. The efficacy of vaccination could be better revealed on experiment with substantially higher number of animals per group (100 animals or more per group) but this scale up experiment was out of financial range of project budget and would rise concerns for ethic committee.
The challenge dose was probably selected well, because in control not vaccinated group some animals developed severe clinical outcomes of infection but others only mild or none. Lower infection dose may be not sufficient to induce clinical signs of infection and may lead to small CFU counts in tissues, thus not enabling to identify whether the vaccine induce protective immune response or not.
- Line 587 “.., CD4+ T-cell responded much better to carrier proteins that to CPS” Probably, CPS is not a good inducer of CD4+ T-cells. Is any information available from other studies about this that you could comment on or compare with your results? It has been described in some articles that CPSs are generally considered poorly immunogenic since they are unable to recruit T cell help for B cell functions. Could please you comment on these papers in your discussion?
Mond, J.J.; Kokai-Kun, J.F. The multifunctional role of antibodies in the protective response to bacterial T cell-independent antigens. In Specialization and Complementation of Humoral Immune Responses to Infection;Manser, T., Ed.; Springer: Berlin/Heidelberg, Germany, 2008; Volume 319, pp. 17–40.
Goyette-Desjardins G, Calzas C, Shiao TC, Neubauer A, Kempker J, Roy R, Gottschalk M, Segura M. Protection against Streptococcus suis Serotype 2 Infection Using a Capsular Polysaccharide Glycoconjugate Vaccine. Infect Immun. 2016 Jun 23;84(7):2059-2075. doi: 10.1128/IAI.00139-16.
Response: We absolutely agree the CPS is very weak inducer of CD4+ T cells and this is the reason why conjugate vaccines are being developed. It is known, that S. suis CPS of serotypes 2 and 14 is not immunogenic (Calzas et al. 2017), while other serotypes could induce some antibody response (Calzas et al. 2015). We improve discussion with respect to your note.
Calzas C, Taillardet M, Fourati IS, Roy D, Gottschalk M, Soudeyns H, Defrance T, Segura M. Evaluation of the Immunomodulatory Properties of Streptococcus suis and Group B Streptococcus Capsular Polysaccharides on the Humoral Response. Pathogens. 2017 Apr 20;6(2):16. doi: 10.3390/pathogens6020016.
Calzas, C., Lemire, P., Auray, G., Gerdts, V., Gottschalk, M., Segura, M. Antibody response specific to the capsular polysaccharide is impaired in Streptococcus suis serotype 2-infected animals. Infect Immun. 2015, 83(1), 441–453. https://doi.org/10.1128/IAI.02427-14
- Line 599 “..the pig from the CPS-OVA group that was euthanized on day five post-infection produced CPS-specific IgG (on an average level of the group) on the day of infection..” Is it known if CPS can generate different types of antibodies against different sugar compositions on CPS and if some of them provide more protection than others? Is any data from other studies about that that could explain the results? It seems that sialic acid in the CPS may interfere with the generation of anti-CPS Ab responses. Could you please comment on this in your paper?
Calzas C, Taillardet M, Fourati IS, Roy D, Gottschalk M, Soudeyns H, Defrance T, Segura M. Evaluation of the Immunomodulatory Properties of Streptococcus suis and Group B Streptococcus Capsular Polysaccharides on the Humoral Response. Pathogens. 2017 Apr 20;6(2):16. doi: 10.3390/pathogens6020016.
Response: Thank you for this note. Yes, it is known, that CPS from different S. suis serotypes having different sugar composition differs in the ability to induce antibody response. This phenomenon was well described in Calzas C, Lemire P, Auray G, Gerdts V, Gottschalk M, Segura M. 2015. Antibody response specific to the capsular polysaccharide is impaired in Streptococcus suis serotype 2-infected animals. Infect Immun 83:441–453. https://doi.org/10.1128/IAI.02427-14.
In article you mentioned (Calzas et al. 2017) was proved, that the sialic acid was essential for the immunogenicity of purified Streptococcus B type III CPS, but its presence/absence had little or no effect on the inability of purified S. suis type 2 CPS to induce a specific Ab response. In the light of this knowledge and because natural S. suis CPS contain sialic acid we do not discuss it further.
- Line 606 “CPS-OVA is promising..” There are several CPS serotypes..how could an ideal vaccine be? A cocktail of the different CPS serotypes or just serotype 2? Could you please comment on this in your discussion?
Response: It is interesting question and very important for further development of vaccines against S. suis infection in pig. There is no information in the literature about protectivity of experimental S. suis vaccines based on one serotype conjugate against infection with other serotypes. As far as we know, the homologous challenge strain was used in all experiments with conjugate vaccines against S. suis in pigs. Based on knowledge obtained with Prevnar vaccine, it is possible that protection of conjugated vaccine against S. suis will be serotype-specific. However, this idea is purely speculative, not supported by any experiment yet. We discussed it in manuscript.
Goyette-Desjardins G, Calzas C, Shiao TC, Neubauer A, Kempker J, Roy R, Gottschalk M, Segura M. 2016. Protection against Streptococcus suis serotype 2 infection using a capsular polysaccharide glycoconjugate vaccine. Infect Immun 84:2059 –2075. doi:10.1128/IAI.00139-16.
Hsueh KJ, Chen MC, Cheng LT, Lee JW, Chung WB, Chu CY. Transcutaneous immunization of Streptococcus suis bacterin using dissolving microneedles. Comp Immunol Microbiol Infect Dis. 2017 Feb;50:78-87. doi: 10.1016/j.cimid.2016.12.001.